# Bayesian Inference for Structured Spike and Slab Priors

**Michael Riis Andersen, Ole Winther & Lars Kai Hansen**
DTU Compute, Technical University of Denmark
DK-2800 Kgs. Lyngby, Denmark
{`miri, olwi, lkh`}@dtu.dk

## Abstract

Sparse signal recovery addresses the problem of solving underdetermined linear inverse problems subject to a sparsity constraint. We propose a novel prior formulation, the structured spike and slab prior, which allows to incorporate a priori knowledge of the sparsity pattern by imposing a spatial Gaussian process on the spike and slab probabilities. Thus, prior information on the structure of the sparsity pattern can be encoded using generic covariance functions. Furthermore, we provide a Bayesian inference scheme for the proposed model based on the expectation propagation framework. Using numerical experiments on synthetic data, we demonstrate the benefits of the model.

## 1   Introduction

Consider a linear inverse problem of the form:

$$\boldsymbol{y} = \boldsymbol{A}\boldsymbol{x} + \boldsymbol{e}, \tag{1}$$

where $\boldsymbol{A} \in \mathbb{R}^{N \times D}$ is the measurement matrix, $\boldsymbol{y} \in \mathbb{R}^N$ is the measurement vector, $\boldsymbol{x} \in \mathbb{R}^D$ is the desired solution and $\boldsymbol{e} \in \mathbb{R}^N$ is a vector of corruptive noise. The field of sparse signal recovery deals with the task of reconstructing the sparse solution $\boldsymbol{x}$ from $(\boldsymbol{A}, \boldsymbol{y})$ in the ill-posed regime where $N < D$. In many applications it is beneficial to encourage a structured sparsity pattern rather than independent sparsity. In this paper we consider a model for exploiting a priori information on the sparsity pattern, which has applications in many different fields, e.g., structured sparse PCA [1], background subtraction [2] and neuroimaging [3].

In the framework of probabilistic modelling sparsity can be enforced using so-called sparsity promoting priors, which conventionally has the following form

$$p(\boldsymbol{x}|\lambda) = \prod_{i=1}^{D} p(x_i|\lambda), \tag{2}$$

where $p(x_i|\lambda)$ is the marginal prior on $x_i$ and $\lambda$ is a fixed hyperparameter controlling the degree of sparsity. Examples of such sparsity promoting priors include the Laplace prior (LASSO [4]), and the Bernoulli-Gaussian prior (the spike and slab model [5]). The main advantage of this formulation is that the inference schemes become relatively simple due to the fact that the prior factorizes over the variables $x_i$. However, this fact also implies that the models cannot encode any prior knowledge of the structure of the sparsity pattern.

One approach to model a richer sparsity structure is the so-called *group sparsity* approach, where the set of variables $\boldsymbol{x}$ has been partitioned into groups beforehand. This

approach has been extensively developed for the $\ell_1$ minimization community, i.e. *group LASSO*, *sparse group LASSO* [6] and *graph LASSO* [7]. Let $\mathcal{G}$ be a partition of the set of variables into $G$ groups. A Bayesian equivalent of group sparsity is the group spike and slab model [8], which takes the form

$$p(\boldsymbol{x}|\boldsymbol{z}) = \prod_{g=1}^{G} \left[ (1 - z_g)\,\delta\left(\boldsymbol{x}_g\right) + z_g \mathcal{N}\left(\boldsymbol{x}_g \middle| 0, \tau \boldsymbol{I}_g\right) \right], \qquad p(\boldsymbol{z}|\boldsymbol{\lambda}) = \prod_{g=1}^{G} \text{Bernoulli}\left(z_g \middle| \lambda_g\right), \quad (3)$$

where $\boldsymbol{z} \in [0,1]^G$ are binary support variables indicating whether the variables in different groups are active or not. Other relevant work includes [9] and [10]. Another more flexible approach is to use a Markov random field (MRF) as prior for the binary variables [2].

Related to the MRF-formulation, we propose a novel model called the *Structured Spike and Slab* model. This model allows us to encode a priori information of the sparsity pattern into the model using generic covariance functions rather than through clique potentials as for the MRF-formulation [2]. Furthermore, we provide a Bayesian inference scheme based on expectation propagation for the proposed model.

## 2   The structured spike and slab prior

We propose a hierarchical prior of the following form:

$$p(\boldsymbol{x}|\boldsymbol{\gamma}) = \prod_{i=1}^{D} p(x_i | g(\gamma_i)), \qquad p(\boldsymbol{\gamma}) = \mathcal{N}\left(\boldsymbol{\gamma}|\boldsymbol{\mu_0}, \boldsymbol{\Sigma_0}\right), \qquad (4)$$

where $g : \mathbb{R} \rightarrow \mathbb{R}$ is a suitable injective transformation. That is, we impose a Gaussian process [11] as a prior on the parameters $\gamma_i$. Using this parametrization, prior knowledge of the structure of the sparsity pattern can be encoded using $\boldsymbol{\mu_0}$ and $\boldsymbol{\Sigma_0}$. The mean value $\boldsymbol{\mu_0}$ controls the prior belief of the support and the covariance matrix determines the prior correlation of the support. In the remainder of this paper we restrict $p(x_i | g(\gamma_i))$ to be a spike and slab model, i.e.

$$p(x_i | z_i) = (1 - z_i)\delta(x_i) + z_i \mathcal{N}\left(x_i | 0, \tau_0\right), \qquad z_i \sim \text{Ber}\left(g(\gamma_i)\right). \qquad (5)$$

This formulation clearly fits into eq. (4) when $z_i$ is marginalized out. Furthermore, we will assume that $g$ is the standard Normal CDF, i.e. $g(x) = \phi(x)$. Using this formulation, the marginal prior probability of the $i$'th weight being active is given by:

$$p(z_i = 1) = \int p(z_i = 1|\gamma_i)p(\gamma_i)\mathrm{d}\gamma_i = \int \phi(\gamma_i)\mathcal{N}\left(\gamma_i|\mu_i, \Sigma_{ii}\right)\mathrm{d}\gamma_i = \phi\left(\frac{\mu_i}{\sqrt{1 + \Sigma_{ii}}}\right). \qquad (6)$$

This implies that the probability of $z_i = 1$ is 0.5 when $\mu_i = 0$ as expected. In contrast to the $\ell_1$-based methods and the MRF-priors, the Gaussian process formulation makes it easy to generate samples from the model. Figures 1(a), 1(b) each show three realizations of the support from the prior using a squared exponential kernel of the form: $\Sigma_{ij} = 50\exp(-\left(i - j\right)^2/2s^2)$ and $\mu_i$ is fixed such that the expected level of sparsity is 10%. It is seen that when the scale, $s$, is small, the support consists of scattered spikes. As the scale increases, the support of the signals becomes more contiguous and clustered, where the sizes of the clusters increase with the scale.

To gain insight into the relationship between $\boldsymbol{\gamma}$ and $\boldsymbol{z}$, we consider the two dimensional system with $\mu_i = 0$ and the following covariance structure

$$\boldsymbol{\Sigma_0} = \kappa \begin{bmatrix} 1 & \rho \\ \rho & 1 \end{bmatrix}, \quad \kappa > 0. \qquad (7)$$

The correlation between $z_1$ and $z_2$ is then computed as a function of $\rho$ and $\kappa$ by sampling. The resulting curves in Figure 1(c) show that the desired correlation is an increasing function of $\rho$ as expected. However, the figure also reveals that for $\rho = 1$, i.e. 100% correlation between the $\gamma$ parameters, does not imply 100% correlation of the support variables $\boldsymbol{z}$. This

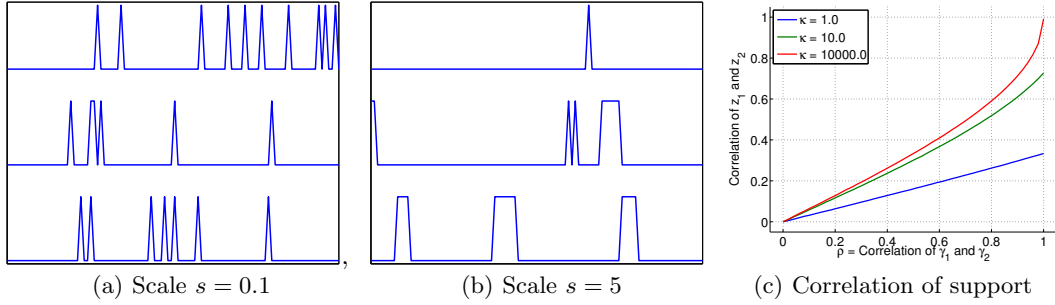

<div align="center">(a) Scale $s = 0.1$      (b) Scale $s = 5$      (c) Correlation of support</div>

Figure 1: (a,b) Realizations of the support $\boldsymbol{z}$ from the prior distribution using a squared exponential covariance function for $\boldsymbol{\gamma}$, i.e. $\Sigma_{ij} = 50 \exp(-(i-j)^2/2s^2)$ and $\boldsymbol{\mu}$ is fixed to match an expected sparsity rate $K/D$ of 10%. (c) Correlation of $z_1$ and $z_2$ as a function of $\rho$ for 5 different values of $A$ obtained by sampling. This prior mean function is fixed at $\mu_i = 0$ for all $i$.

is due to the fact that there are two levels of uncertainty in the prior distribution of the support. That is, first we sample $\boldsymbol{\gamma}$, and then we sample the support $\boldsymbol{z}$ conditioned on $\boldsymbol{\gamma}$.

The proposed prior formulation extends easily to the multiple measurement vector (MMV) formulation [12, 13, 14], in which multiple linear inverse problems are solved simultaneously. The most straightforward way is to assume all problem instances share the same support variable, commonly known as joint sparsity [14]

$$p\left(\boldsymbol{X}|\boldsymbol{z}\right) = \prod_{t=1}^{T}\prod_{i=1}^{D}\left[(1-z_i)\delta(x_i^t) + z_i\mathcal{N}\left(x_i^t|0,\tau\right)\right], \tag{8}$$

$$p(z_i|\gamma_i) = \text{Ber}\left(z_i|\phi(\gamma_i)\right), \tag{9}$$

$$p(\boldsymbol{\gamma}) = \mathcal{N}\left(\boldsymbol{\gamma}|\boldsymbol{\mu}_0, \boldsymbol{\Sigma}_0\right), \tag{10}$$

where $\boldsymbol{X} = \begin{bmatrix} \boldsymbol{x}^1 & \dots & \boldsymbol{x}^T \end{bmatrix} \in \mathbb{R}^{D \times T}$. The model can also be extended to problems, where the sparsity pattern changes in time

$$p\left(\boldsymbol{X}|\boldsymbol{z}\right) = \prod_{t=1}^{T}\prod_{i=1}^{D}\left[(1-z_i^t)\delta(x_i^t) + z_i^t\mathcal{N}\left(x_i^t|0,\tau\right)\right], \tag{11}$$

$$p(z_i^t|\gamma_i^t) = \text{Ber}\left(z_i^t|\phi(\gamma_i^t)\right), \tag{12}$$

$$p(\boldsymbol{\gamma}_1, ..., \boldsymbol{\gamma}_T) = \mathcal{N}\left(\boldsymbol{\gamma}_1|\boldsymbol{\mu}_0, \boldsymbol{\Sigma}_0\right)\prod_{t=2}^{T}\mathcal{N}\left(\boldsymbol{\gamma}_t|(1-\alpha)\boldsymbol{\mu}_0 + \alpha\boldsymbol{\gamma}_{t-1}, \beta\boldsymbol{\Sigma}_0\right), \tag{13}$$

where the parameters $0 \le \alpha \le 1$ and $\beta \ge 0$ controls the temporal dynamics of the support.

## 3   Bayesian inference using expectation propagation

In this section we combine the structured spike and slab prior as given in eq. (5) with an isotropic Gaussian noise model and derive an inference algorithm based on expectation propagation. The likelihood function is $p(\boldsymbol{y}|\boldsymbol{x}) = \mathcal{N}\left(\boldsymbol{y}|\boldsymbol{Ax}, \sigma_0^2\boldsymbol{I}\right)$ and the joint posterior distribution of interest thus becomes

$$p(\boldsymbol{x}, \boldsymbol{z}, \boldsymbol{\gamma}|\boldsymbol{y}) = \frac{1}{Z}p(\boldsymbol{y}|\boldsymbol{x})p(\boldsymbol{x}|\boldsymbol{z})p(\boldsymbol{z}|\boldsymbol{\gamma})p(\boldsymbol{\gamma}) \tag{14}$$

$$= \frac{1}{Z}\underbrace{\mathcal{N}\left(\boldsymbol{y}|\boldsymbol{Ax}, \sigma_0^2\boldsymbol{I}\right)}_{f_1}\underbrace{\prod_{i=1}^{D}\left[(1-z_i)\delta(x_i) + z_i\mathcal{N}\left(x_i|0,\tau_0\right)\right]}_{f_2}\underbrace{\prod_{i=1}^{D}\text{Ber}\left(z_i|\phi\left(\gamma_i\right)\right)}_{f_3}\underbrace{\mathcal{N}\left(\boldsymbol{\gamma}|\boldsymbol{\mu}_0, \boldsymbol{\Sigma}_0\right)}_{f_4},$$

<div align="center">3</div>

where $Z$ is the normalization constant independent of $\boldsymbol{x}, \boldsymbol{z}$ and $\boldsymbol{\gamma}$. Unfortunately, the true posterior is intractable and therefore we have to settle for an approximation. In particular, we apply the framework of expectation propagation (EP) [15, 16], which is an iterative deterministic framework for approximating probability distributions using distributions from the exponential family. The algorithm proposed here can be seen as an extension of the work in [8].

As shown in eq. (14), the true posterior is a composition of 4 factors, i.e. $f_a$ for $a = 1, .., 4$. The terms $f_2$ and $f_3$ are further decomposed into $D$ conditionally independent factors

$$f_2(\boldsymbol{x}, \boldsymbol{z}) = \prod_{i=1}^{D} f_{2,i}(x_i, z_i) = \prod_{i=1}^{D} \left[ (1 - z_i)\delta(x_i) + z_i \mathcal{N}\left(x_i | 0, \tau_0\right) \right], \tag{15}$$

$$f_3(\boldsymbol{z}, \boldsymbol{\gamma}) = \prod_{i=1}^{D} f_{3,i}(z_i, \gamma_i) = \prod_{i=1}^{D} \text{Ber}\left(z_i | \phi\left(\gamma_i\right)\right) \tag{16}$$

The idea is then to approximate each term in the true posterior density, i.e. $f_a$, by simpler terms, i.e. $\tilde{f}_a$ for $a = 1, .., 4$. The resulting approximation $Q(\boldsymbol{x}, \boldsymbol{z}, \boldsymbol{\gamma})$ then becomes

$$Q(\boldsymbol{x}, \boldsymbol{z}, \boldsymbol{\gamma}) = \frac{1}{Z_{EP}} \prod_{a=1}^{4} \tilde{f}_a(\boldsymbol{x}, \boldsymbol{z}, \boldsymbol{\gamma}). \tag{17}$$

The terms $\tilde{f}_1$ and $\tilde{f}_4$ can be computed exact. In fact, $\tilde{f}_4$ is simply equal to the prior over $\boldsymbol{\gamma}$ and $\tilde{f}_1$ is a multivariate Gaussian distribution with mean $\tilde{\boldsymbol{m}}_1$ and covariance matrix $\tilde{\boldsymbol{V}}_1$ determined by $\tilde{\boldsymbol{V}}_1^{-1}\tilde{\boldsymbol{m}}_1 = \frac{1}{\sigma^2}\boldsymbol{A}^T\boldsymbol{y}$ and $\tilde{\boldsymbol{V}}_1^{-1} = \frac{1}{\sigma^2}\boldsymbol{A}^T\boldsymbol{A}$. Therefore, we only have to approximate the factors $\tilde{f}_2$ and $\tilde{f}_3$ using EP. Note that the exact term $f_1$ is a distribution of $\boldsymbol{y}$ conditioned on $\boldsymbol{x}$, whereas the approximate term $\tilde{f}_1$ is a function of $\boldsymbol{x}$ that depends on $\boldsymbol{y}$ through $\tilde{\boldsymbol{m}}_1$ and $\tilde{\boldsymbol{V}}_1$ etc. In order to take full advantage of the structure of the true posterior distribution, we will further assume that the terms $\tilde{f}_2$ and $\tilde{f}_3$ also are decomposed into $D$ independent factors.

The EP scheme provides great flexibility in the choice of the approximating factors. This choice is a trade-off between analytical tractability and sufficient flexibility for capturing the important characteristics of the true density. Due to the product over the binary support variables $\{z_i\}$ for $i = 1, .., D$, the true density is highly multimodal. Finally, $f_2$ couples the variables $\boldsymbol{x}$ and $\boldsymbol{z}$, while $f_3$ couples the variables $\boldsymbol{z}$ and $\boldsymbol{\gamma}$. Based on these observations, we choose $\tilde{f}_2$ and $\tilde{f}_3$ to have the following forms

$$\tilde{f}_2(\boldsymbol{x}, \boldsymbol{z}) \propto \prod_{i=1}^{D} \mathcal{N}\left(x_i | \tilde{m}_{2,i}, \tilde{v}_{2,i}\right) \prod_{i=1}^{D} \text{Ber}\left(z_i | \phi\left(\tilde{\gamma}_{2,i}\right)\right) = \mathcal{N}\left(\boldsymbol{x} | \tilde{\boldsymbol{m}}_2, \tilde{\boldsymbol{V}}_2\right) \prod_{i=1}^{D} \text{Ber}\left(z_i | \phi\left(\tilde{\gamma}_{2,i}\right)\right),$$

$$\tilde{f}_3(\boldsymbol{z}, \boldsymbol{\gamma}) \propto \prod_{i=1}^{D} \text{Ber}\left(z_i | \phi\left(\tilde{\gamma}_{3,i}\right)\right) \prod_{i=1}^{D} \mathcal{N}\left(\gamma_i | \tilde{\mu}_{3,i}, \tilde{\sigma}_{3,i}\right) = \mathcal{N}\left(\boldsymbol{\gamma} | \tilde{\boldsymbol{\mu}}_3, \tilde{\boldsymbol{\Sigma}}_3\right) \prod_{i=1}^{D} \text{Ber}\left(z_i | \phi\left(\tilde{\gamma}_{2,i}\right)\right),$$

where $\tilde{\boldsymbol{m}}_2 = [\tilde{m}_{2,1}, .., \tilde{m}_{2,D}]^T$, $\tilde{\boldsymbol{V}}_2 = \text{diag}\left(\tilde{v}_{2,1}, ..., \tilde{v}_{2,D}\right)$ and analogously for $\tilde{\boldsymbol{\mu}}_3$ and $\tilde{\boldsymbol{\Sigma}}_3$. These choices lead to a joint variational approximation $Q(\boldsymbol{x}, \boldsymbol{z}, \boldsymbol{\gamma})$ of the form

$$Q(\boldsymbol{x}, \boldsymbol{z}, \boldsymbol{\gamma}) = \mathcal{N}\left(\boldsymbol{x} | \tilde{\boldsymbol{m}}, \tilde{\boldsymbol{V}}\right) \prod_{i=1}^{D} \text{Ber}\left(z_i | g\left(\tilde{\gamma}_i\right)\right) \mathcal{N}\left(\boldsymbol{\gamma} | \tilde{\boldsymbol{\mu}}, \tilde{\boldsymbol{\Sigma}}\right), \tag{18}$$

where the joint parameters are given by

$$\tilde{\boldsymbol{V}} = \left(\tilde{\boldsymbol{V}}_1^{-1} + \tilde{\boldsymbol{V}}_2^{-1}\right)^{-1}, \qquad \tilde{\boldsymbol{m}} = \tilde{\boldsymbol{V}}\left(\tilde{\boldsymbol{V}}_1^{-1}\tilde{\boldsymbol{m}}_1 + \tilde{\boldsymbol{V}}_2^{-1}\tilde{\boldsymbol{m}}_2\right) \tag{19}$$

$$\tilde{\boldsymbol{\Sigma}} = \left(\tilde{\boldsymbol{\Sigma}}_3^{-1} + \tilde{\boldsymbol{\Sigma}}_4^{-1}\right)^{-1}, \qquad \tilde{\boldsymbol{\mu}} = \tilde{\boldsymbol{\Sigma}}\left(\tilde{\boldsymbol{\Sigma}}_3^{-1}\tilde{\boldsymbol{\mu}}_3 + \tilde{\boldsymbol{\Sigma}}_4^{-1}\tilde{\boldsymbol{\mu}}_4\right) \tag{20}$$

$$\tilde{\gamma}_j = \phi^{-1}\left[\left(\frac{(1 - \phi(\tilde{\gamma}_{2,j}))(1 - \phi(\tilde{\gamma}_{3,j}))}{\phi(\tilde{\gamma}_{2,j})\phi(\tilde{\gamma}_{3,j})} + 1\right)^{-1}\right], \quad \forall j \in \{1, .., D\}. \tag{21}$$

where $\phi^{-1}(x)$ is the probit function. The function in eq. (21) amounts to computing the product of two Bernoulli densities parametrized using $\phi(\cdot)$.

- • Initialize approximation terms $\tilde{f}_a$ for $a = 1, 2, 3, 4$ and $Q$
- • Repeat until stopping criteria
  - – For each $\tilde{f}_{2,i}$:
    - ∗ Compute cavity distribution: $Q^{\backslash 2,i} \propto \frac{Q}{\tilde{f}_{2,i}}$
    - ∗ Minimize: $\mathrm{KL}\big(f_{2,i}Q^{\backslash 2,i}\big|\big|Q^{2,\mathrm{new}}\big)$ w.r.t. $Q^{\mathrm{new}}$
    - ∗ Compute: $\tilde{f}_{2,i} \propto \frac{Q^{2,\mathrm{new}}}{Q^{\backslash 2,i}}$ to update parameters $\tilde{m}_{2,i}, \tilde{v}_{2,i}$ and $\tilde{\gamma}_{2,i}$.
  - – Update joint approximation parameters: $\tilde{\boldsymbol{m}}, \tilde{\boldsymbol{V}}$ and $\tilde{\boldsymbol{\gamma}}$
  - – For each $\tilde{f}_{3,i}$:
    - ∗ Compute cavity distribution: $Q^{\backslash 3,i} \propto \frac{Q}{\tilde{f}_{3,i}}$
    - ∗ Minimize: $\mathrm{KL}\big(f_{3,i}Q^{\backslash 3,i}\big|\big|Q^{3,\mathrm{new}}\big)$ w.r.t. $Q^{\mathrm{new}}$
    - ∗ Compute: $\tilde{f}_{3,i} \propto \frac{Q^{3,\mathrm{new}}}{Q^{\backslash 3,i}}$ to update parameters $\tilde{\mu}_{3,i}, \tilde{\sigma}_{3,i}$ and $\tilde{\gamma}_{3,i}$
  - – Update joint approximation parameters: $\tilde{\boldsymbol{\mu}}, \tilde{\boldsymbol{\Sigma}}$ and $\tilde{\boldsymbol{\gamma}}$

Figure 2: Proposed algorithm for approximating the joint posterior distribution over $\boldsymbol{x}, \boldsymbol{z}$ and $\boldsymbol{\gamma}$.

## 3.1 The EP algorithm

Consider the update of the term $\tilde{f}_{a,i}$ for a given $a$ and a given $i$, where $\tilde{f}_a = \prod_i \tilde{f}_{a,i}$. This update is performed by first removing the contribution of $\tilde{f}_{a,i}$ from the joint approximation by forming the so-called cavity distribution

$$Q^{\backslash a,i} \propto \frac{Q}{\tilde{f}_{a,i}} \tag{22}$$

followed by the minimization of the Kullbach-Leibler [17] divergence between $f_{a,i}Q^{\backslash a,i}$ and $Q^{a,\mathrm{new}}$ w.r.t. $Q^{a,\mathrm{new}}$. For distributions within the exponential family, minimizing this form of KL divergence amounts to matching moments between $f_{a,i}Q^{\backslash 2,i}$ and $Q^{a,\mathrm{new}}$ [15]. Finally, the new update of $\tilde{f}_{a,i}$ is given by

$$\tilde{f}_{a,i} \propto \frac{Q^{a,\mathrm{new}}}{Q^{\backslash a,i}}. \tag{23}$$

After all the individual approximation terms $\tilde{f}_{a,i}$ for $a = 1, 2$ and $i = 1, .., D$ have been updated, the joint approximation is updated using eq. (19)-(21). To minimize the computational load, we use parallel updates of $\tilde{f}_{2,i}$ [8] followed by parallel updates of $\tilde{f}_{3,i}$ rather than the conventional sequential update scheme. Furthermore, due to the fact that $\tilde{f}_2$ and $\tilde{f}_3$ factorizes, we only need the marginals of the cavity distributions $Q^{\backslash a,i}$ and the marginals of the updated joint distributions $Q^{a,\mathrm{new}}$ for $a = 2, 3$.

Computing the cavity distributions and matching the moments are tedious, but straightforward. The moments of $f_{a,i}Q^{\backslash 2,i}$ require evaluation of the zeroth, first and second order moment of the distributions of the form $\phi(\gamma_i)\mathcal{N}\big(\gamma_i\big|\mu_i, \Sigma_{ii}\big)$. Derivation of analytical expressions for these moments can be found in [11]. See the supplementary material for more details. The proposed algorithm is summarized in figure 2. Note, that the EP framework also provides an approximation of the marginal likelihood [11], which can be useful for learning the hyperparameters of the model. Furthermore, the proposed inference scheme can easily be extended to the MMV formulation eq. (8)-(10) by introducing a $\tilde{f}_{2,i}^t$ for each time step $t = 1, .., T$.

## 3.2 Computational details

Most linear inverse problems of practical interest are high dimensional, i.e. $D$ is large. It is therefore of interest to simplify the computational complexity of the algorithm as much as possible. The dominating operations in this algorithm are the inversions of the two $D \times D$ covariance matrices in eq. (19) and eq. (20), and therefore the algorithm scales as $\mathcal{O}\left(D^3\right)$. But $\tilde{\boldsymbol{V}}_1$ has low rank and $\tilde{\boldsymbol{V}}_2$ is diagonal, and therefore we can apply the Woodbury matrix identity [18] to eq. (19) to get

$$\tilde{\boldsymbol{V}} = \tilde{\boldsymbol{V}}_2 - \tilde{\boldsymbol{V}}_2 \boldsymbol{A}^T \left(\sigma_o^2 \boldsymbol{I} + \boldsymbol{A}\tilde{\boldsymbol{V}}_2 \boldsymbol{A}^T\right)^{-1} \boldsymbol{A}\tilde{\boldsymbol{V}}_2. \tag{24}$$

For $N < D$, this scales as $\mathcal{O}\left(ND^2\right)$, where $N$ is the number of observations. Unfortunately, we cannot apply the same identity to the inversion in eq. (20) since $\tilde{\boldsymbol{\Sigma}}_4$ has full rank and is non-diagonal in general. The eigenvalue spectrum of many prior covariance structures of interest, i.e. simple neighbourhoods etc., decay relatively fast. Therefore, we can approximate $\boldsymbol{\Sigma}_0$ with a low rank approximation $\boldsymbol{\Sigma}_0 \approx \boldsymbol{P}\Lambda\boldsymbol{P}^T$, where $\Lambda \in \mathbb{R}^{R \times R}$ is a diagonal matrix of the $R$ largest eigenvalues and $\boldsymbol{P} \in \mathbb{R}^{D \times R}$ is the corresponding eigenvectors. Using the R-rank approximation, we can now invoke the Woodbury matrix identity again to get:

$$\tilde{\boldsymbol{\Sigma}} = \tilde{\boldsymbol{\Sigma}}_3 + \tilde{\boldsymbol{\Sigma}}_3 \boldsymbol{P}\left(\Lambda + \boldsymbol{P}^T \tilde{\boldsymbol{\Sigma}}_3 \boldsymbol{P}\right)^{-1} \boldsymbol{P}^T \tilde{\boldsymbol{\Sigma}}_3. \tag{25}$$

Similarly, for $R < D$, this scales as $\mathcal{O}\left(RD^2\right)$. Another better approach that preserves the total variance would be to use probabilistic PCA [19] to approximate $\boldsymbol{\Sigma}_0$. A third alternative is to consider other structures for $\boldsymbol{\Sigma}_0$, which facilitate fast matrix inversions such as block structures and Toeplitz structures. Numerical issues can arise in EP implementations and in order to avoid this, we use the same precautions as described in [8].

## 4 Numerical experiments

This section describes a series of numerical experiments that have been designed and conducted in order to investigate the properties of the proposed algorithm.

### 4.1 Experiment 1

The first experiment compares the proposed method to the LARS algorithm [20] and to the BG-AMP method [21], which is an approximate message passing-based method for the spike and slab model. We also compare the method to an "oracle least squares estimator" that knows the true support of the solutions. We generate 100 problem instances from $\boldsymbol{y} = \boldsymbol{A}\boldsymbol{x}_0 + \boldsymbol{e}$, where the solutions vectors have been sampled from the proposed prior using the kernel $\Sigma_{i,j} = 50\exp(-||i - j||_2^2/(2 \cdot 10^2))$, but constrained to have a fixed sparsity level of the $K/D = 0.25$. That is, each solution $\boldsymbol{x}_0$ has the same number of non-zero entries, but different sparsity patterns. We vary the degree of undersampling from $N/D = 0.05$ to $N/D = 0.95$. The elements of $\boldsymbol{A} \in \mathbb{R}^{N \times 250}$ are i.i.d Gaussian and the columns of $\boldsymbol{A}$ have been scaled to unit $\ell_2$-norm. The SNR is fixed at 20dB. We apply the four methods to each of the 100 problems, and for each solution we compute the Normalized Mean Square Error (NMSE) between the true signal $\boldsymbol{x}_0$ and the estimated signal $\hat{\boldsymbol{x}}$ as well as the $F$-measure:

$$\text{NMSE} = \frac{||\boldsymbol{x}_0 - \hat{\boldsymbol{x}}||_2}{||\boldsymbol{x}_0||_2} \qquad\qquad F = 2\frac{\text{precision} \cdot \text{recall}}{\text{precision} + \text{recall}}, \tag{26}$$

where precision and recall are computed using a MAP estimate of the support. For the structured spike and slab method, we consider three different covariance structures: $\Sigma_{ij} = \kappa \cdot \delta(i - j)$, $\Sigma_{ij} = \kappa \exp(-||i - j||_2/s)$ and $\Sigma_{ij} = \kappa \exp(-||i - j||_2^2/(2s^2))$ with parameters $\kappa = 50$ and $s = 10$. In each case, we use a $R = 50$ rank approximation of $\Sigma$. The average results are shown in figures 3(a)-(f). Figure (a) shows an example of one of the sampled vectors $\boldsymbol{x}_0$ and figure (b) shows the three covariance functions.

From figure 3(c)-(d), it is seen that the two EP methods with neighbour correlation are able to improve the phase transition point. That is, in order to obtain a reconstruction

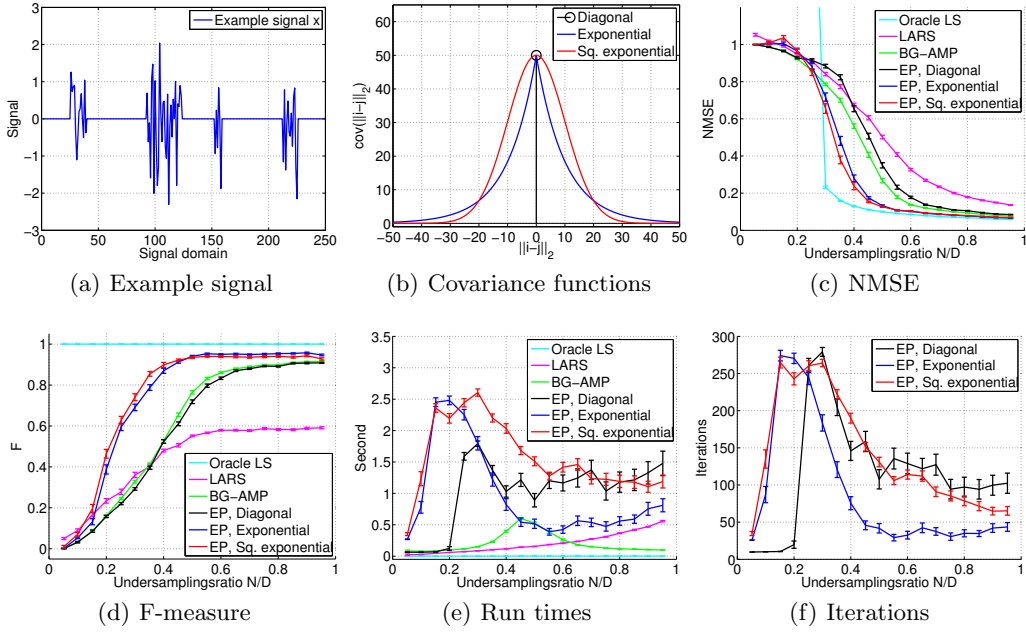

(a) Example signal        (b) Covariance functions        (c) NMSE

(d) F-measure        (e) Run times        (f) Iterations

Figure 3: Illustration of the benefit of modelling the additional structure of the sparsity pattern. 100 problem instances are generated using the linear measurement model $\boldsymbol{y} = \boldsymbol{A}\boldsymbol{x} + \boldsymbol{e}$, where elements of $\boldsymbol{A} \in \mathbb{R}^{N \times 250}$ are i.i.d Gaussian and the columns are scaled to unit $\ell_2$-norm. The solutions $\boldsymbol{x}_0$ are sampled from the prior in eq. (5) with hyperparameters $\Sigma_{ij} = 50 \exp\left[-\|i-j\|_2 / \left(2 \cdot 10^2\right)\right]$ and a fixed level of sparsity of $K/D = 0.25$. For EP methods, the $\boldsymbol{\Sigma}_0$ matrix is approximated using a rank 50 matrix. SNR is fixed at 20dB.

of the signal such that $F \approx 0.8$, EP with diagonal covariance and BG-AMP need an undersamplingratio of $N/D \approx 0.55$, while the EP methods with neighbour correlation only need $N/D \approx 0.35$ to achieve $F \approx 0.8$. For this specific problem, this means that utilizing the neighbourhood structure allows us to reconstruct the signal with 50 fewer observations. Note that, the reconstruction using the exponential covariance function does also improve the result even if the true underlying covariance structure corresponds to a squared exponential function. Furthermore, we see similar performance of BG-AMP and EP with a diagonal covariance matrix. This is expected for problems where $A_{ij}$ is drawn iid as assumed in BG-AMP. However, the price of the improved phase transition is clear from figure 3(e). The proposed algorithm has significantly higher computational complexity than BG-AMP and LARS. Figure 4(a) shows the posterior mean of $\boldsymbol{z}$ for the signal shown in figure 3(a). Here it is seen that the two models with neighbour correlation provide a better approximation to the posterior activation probabilities. Figure 4(b) shows the posterior mean of $\boldsymbol{\gamma}$ for the model with the squared exponential kernel along with $\pm$ one standard deviation.

## 4.2   Experiment 2

In this experiment we consider an application of the MMV formulation as given in eq. (8)-(10), namely EEG source localization with synthetic sources [22]. Here we are interested in localizing the active sources within a specific region of interest on the cortical surface (grey area on figure 5(a)). To do this, we now generate a problem instance of $\boldsymbol{Y} = \boldsymbol{A}_{\mathrm{EEG}}\boldsymbol{X}_0 + \boldsymbol{E}$ using the procedure as described in experiment 1, where $\boldsymbol{A}_{\mathrm{EEG}} \in \mathbb{R}^{128 \times 800}$ is now a submatrix of a real EEG forward matrix corresponding to the grey area on the figure. The condition number of $\boldsymbol{A}_{\mathrm{EEG}}$ is $\approx 8 \cdot 10^{15}$. The true sources $\boldsymbol{X}_0 \in \mathbb{R}^{800 \times 20}$ are sampled from the structured spike and slab prior in eq. (8) using a squared exponential kernel with parameters $A = 50$, $s = 10$ and $T = 20$. The number of active sources is 46, i.e. $\boldsymbol{x}$ has 46 non-zero rows. SNR is fixed to 20dB. The true sources are shown in figure 5(a). We now use the EP algorithm to recover the sources using the true prior, i.e. squared exponential kernel and

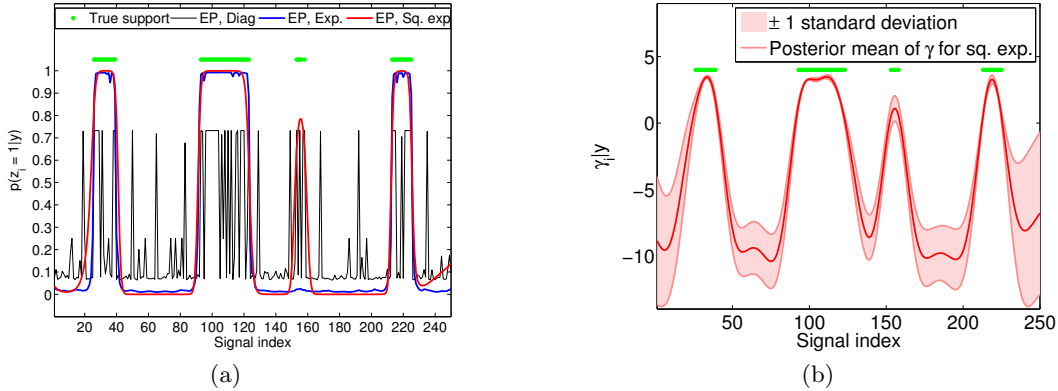

(a)                                          (b)

Figure 4: (a) Marginal posterior means over $\boldsymbol{z}$ obtained using the structured spike and slab model for the signal in figure 3(a). The experiment set-up is the as described in figure 3, except the undersamplingsratio is fixed to $N/D = 0.5$. (b) The posterior mean of $\boldsymbol{\gamma}$ superimposed with $\pm$ one standard deviation. The green dots indicate the true support.

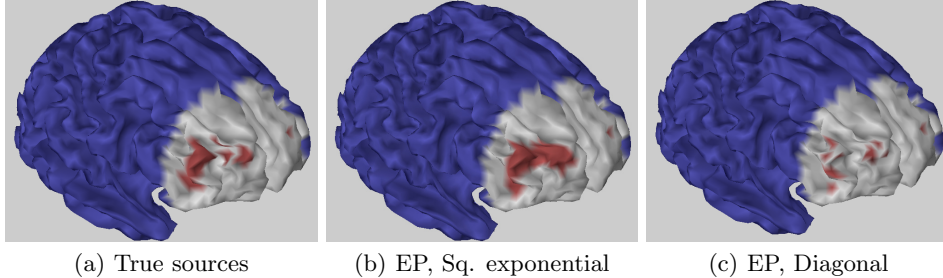

(a) True sources          (b) EP, Sq. exponential          (c) EP, Diagonal

Figure 5: Source localization using synthetic sources. The $\boldsymbol{A} \in \mathbb{R}^{128 \times 800}$ is a submatrix (grey area) of a real EEG forward matrix. (a) True sources. (b) Reconstruction using the true prior , $F_{sq} = 0.78$. (c) Reconstruction using a diagonal covariance matrix, $F_{\text{diag}} = 0.34$.

the results are shown in figure 5(b). We see that the algorithm detects most of the sources correctly, even the small blob on the right hand side. However, it also introduces a small number of false positives in the neighbourhood of the true active sources. The resulting $F$-measure is $F_{sq} = 0.78$. Figure 5(c) shows the result of reconstructing the sources using a diagonal covariance matrix, where $F_{\text{diag}} = 0.34$. Here the BG-AMP algorithm is expected to perform poorly due to the heavy violation of the assumption of $A_{ij}$ being Gaussian iid.

### 4.3 Experiment 3

We have also recreated the Shepp-Logan Phantom experiment from [2] with $D = 10^4$ unknowns, $K = 1723$ non-zero weights, $N = 2K$ observations and SNR $= 10dB$ (see supplementary material for more details). The EP method yields $F_{sq} = 0.994$ and NMSE$_{sq}$ $= 0.336$ for this experiment, whereas BG-AMP yields $F = 0.624$ and NMSE $= 0.717$. For reference, the oracle estimator yields NMSE $= 0.326$.

## 5 Conclusion and outlook

We introduced the structured spike and slab model, which allows incorporation of a priori knowledge of the sparsity pattern. We developed an expectation propagation-based algorithm for Bayesian inference under the proposed model. Future work includes developing a scheme for learning the structure of the sparsity pattern and extending the algorithm to the multiple measurement vector formulation with slowly changing support.

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
