[Supplementary Material]

# Supplementary Material: Bayesian Inference for Structured Spike and Slab Priors

**Michael Riis Andersen, Ole Winther & Lars Kai Hansen**
DTU Compute, Technical University of Denmark
DK-2800 Kgs. Lyngby, Denmark
{miri, olwi, lkh}@dtu.dk

The purpose of this supplementary document is to provide further details for the paper "Bayesian Inference for Structured Spike and Slab Priors".

## Model specification

The model is a standard linear model of the form

$$\boldsymbol{y} = \boldsymbol{A}\boldsymbol{x} + \boldsymbol{e} \tag{1}$$

where $\boldsymbol{y} \in \mathbb{R}^N, \boldsymbol{x} \in \mathbb{R}^D, \boldsymbol{A} \in \mathbb{R}^{N \times D}$ and $\boldsymbol{e}\mathbb{R}^N$. The noise in the system is assumed to be i.i.d. Gaussian distributed

$$p(\boldsymbol{y}|\boldsymbol{x}) = \mathcal{N}\left(\boldsymbol{y}|\boldsymbol{A}\boldsymbol{x}, \sigma_0^2 \boldsymbol{I}\right) \tag{2}$$

and we impose a spike and slab model on the prior

$$p(x_i|z_i) = (1 - z_i)\,\delta\left(x_i\right) + z_i \mathcal{N}\left(x_i|0, \tau\right) \tag{3}$$

The support variables $\boldsymbol{z} = \{z_1, z_2, ..., z_D\}$ is assumed to be Bernoulli distribution

$$p(z_i|\gamma_i) = \mathrm{Ber}\left(z_i|\phi\left(\gamma_i\right)\right) \tag{4}$$

where $\phi : \mathbb{R} \to (0, 1)$ is the standard normal CDF function. Finally, we impose a multivariate Gaussian density on $\boldsymbol{\gamma} = \{\gamma_1, \gamma_2, ..., \gamma_D\}$

$$p(\boldsymbol{\gamma}) = \mathcal{N}\left(\boldsymbol{\gamma}|\boldsymbol{\mu}_0, \boldsymbol{\Sigma}_0\right) \tag{5}$$

The joint posterior over $\boldsymbol{x}, \boldsymbol{z}$ and $\boldsymbol{\gamma}$

$$p\left(\boldsymbol{x}, \boldsymbol{z}, \boldsymbol{\gamma}|\boldsymbol{y}\right) = \frac{1}{Z}\mathcal{N}\left(\boldsymbol{y}|\boldsymbol{A}\boldsymbol{x}, \sigma_0^2 \boldsymbol{I}\right)\prod_{i=1}^{D}\left[(1 - z_i)\,\delta\left(x_i\right) + z_i \mathcal{N}\left(x_i|0, \tau\right)\right]\prod_{i=1}^{D}\mathrm{Ber}\left(z_i|\phi\left(\gamma_i\right)\right)\mathcal{N}\left(\boldsymbol{\gamma}|\boldsymbol{\mu}_0, \boldsymbol{\Sigma}_0\right) \tag{6}$$

## Variational distribution for Expectation propagation

The joint variational distribution is of the form

$$Q\left(\boldsymbol{x}, \boldsymbol{z}, \boldsymbol{\gamma}\right) = \mathcal{N}\left(\boldsymbol{x}|\tilde{\boldsymbol{m}}, \tilde{\boldsymbol{V}}\right)\prod_{i=1}^{D}\mathrm{Ber}\left(z_i|\phi\left(\hat{\gamma}_i\right)\right)\mathcal{N}\left(\boldsymbol{\gamma}|\tilde{\boldsymbol{\mu}}, \tilde{\boldsymbol{\Sigma}}\right)$$

$$= \tilde{f}_1\left(\boldsymbol{x}\right)\tilde{f}_2\left(\boldsymbol{z}\right)\tilde{f}_3\left(\boldsymbol{z}, \boldsymbol{\gamma}\right)\tilde{f}_4\left(\boldsymbol{\gamma}\right)$$

$$= \underbrace{\mathcal{N}\left(\boldsymbol{x}|\tilde{\boldsymbol{m}}_1, \tilde{\boldsymbol{V}}_1\right)}_{\tilde{f}_1}\underbrace{\mathcal{N}\left(\boldsymbol{x}|\tilde{\boldsymbol{m}}_2, \tilde{\boldsymbol{V}}_2\right)\prod_{i=1}^{D}\mathrm{Ber}\left(z_i|\phi\left(\tilde{\gamma}_{2,i}\right)\right)}_{\tilde{f}_2}\underbrace{\prod_{i=1}^{D}\mathrm{Ber}\left(z_i|\phi\left(\tilde{\gamma}_{3,i}\right)\right)\mathcal{N}\left(\boldsymbol{\gamma}|\tilde{\boldsymbol{\mu}}_3, \tilde{\boldsymbol{\Sigma}}_3\right)}_{\tilde{f}_3}\underbrace{\mathcal{N}\left(\boldsymbol{\gamma}|\tilde{\boldsymbol{\mu}}_4, \tilde{\boldsymbol{\Sigma}}_4\right)}_{\tilde{f}_4}$$

$$\tag{7}$$

where $\tilde{V}_2$ is diagonal with elements $\{\tilde{v}_{2,j}\}$ and similar for $\tilde{\Sigma}_3$. We immediately make the following identifications. The first approximation term $\tilde{f}_1$ corresponds to the likelihood term and the fourth term $\tilde{f}_4$ corresponds to the prior on $\gamma$. Thus, we only have to approximate the parameters in the second and third term, i.e. $\tilde{f}_2$ and $\tilde{f}_3$ .

**Computing joint approximation Q from $\tilde{f}_a$**

Given $\tilde{f}_i$ for $i = 1, 2, 3, 4$, we get:

$$\tilde{V} = \left( \tilde{V}_1^{-1} + \tilde{V}_2^{-1} \right)^{-1} \tag{8}$$

$$\tilde{m} = \left( \tilde{V}_1^{-1} \tilde{m}_1 + \tilde{V}_2^{-1} \tilde{m}_2 \right) \tag{9}$$

$$\tilde{\Sigma} = \left( \tilde{\Sigma}_3^{-1} + \tilde{\Sigma}_4^{-1} \right)^{-1} \tag{10}$$

$$\tilde{\mu} = \tilde{\Sigma} \left( \tilde{\Sigma}_3^{-1} \tilde{\mu}_3 + \tilde{\Sigma}_4^{-1} \tilde{\mu}_4 \right) \tag{11}$$

$$\tilde{\gamma}_j = t\left( \tilde{\gamma}_{2,j}, \tilde{\gamma}_{3,j} \right), \quad \forall j \in \{1, .., D\} \tag{12}$$

We have defined the following auxilary functions

$$d\left(x, y\right) = \phi^{-1}\left[\left(\frac{(1 - \phi(x))\,\phi(y)}{(1 - \phi(y))\,\phi(x)} + 1\right)^{-1}\right], \quad t\left(x, y\right) = \phi^{-1}\left[\left(\frac{(1 - \phi(x))\,(1 - \phi(y))}{\phi(x)\phi(y)} + 1\right)^{-1}\right]$$

where $\phi^{-1}(x)$ is the probit function. The function $t(\cdot, \cdot)$ amounts to computing the product of two Bernoulli densities parametrized using $\phi(\cdot)$ and $d(\cdot, \cdot)$ is the corresponding function for quotients of Bernoulli densities.

**Computing the cavity distributions for $\tilde{f}_2$**

$$Q^{\backslash 2,j}(\boldsymbol{x}, \boldsymbol{z}, \boldsymbol{\gamma}) = \frac{Q(\boldsymbol{x}, \boldsymbol{z}, \boldsymbol{\gamma})}{f_{2,j}(x_j, z_j)} = \frac{\mathcal{N}\left(\boldsymbol{x} | \tilde{m}, \tilde{V}\right) \prod_{i=1}^{D} \mathrm{Ber}\left(z_i | \phi\left(\tilde{\gamma}_i\right)\right) \mathcal{N}\left(\tilde{\gamma} | \tilde{\mu}, \tilde{\Sigma}\right)}{\mathrm{Ber}\left(z_j | \phi\left(\tilde{\gamma}_{2,j}\right)\right) \mathcal{N}\left(x_j | \tilde{m}_{2,j}, \tilde{v}_{2,j}\right)} \tag{13}$$

The parameters for the $j$'th marginal cavity distribution then becomes:

$$v_j^{\backslash 2,j} = \left( \left(\tilde{V}_{jj}\right)^{-1} - \tilde{v}_{2,j}^{-1} \right)^{-1} \tag{14}$$

$$m_j^{\backslash 2,j} = v_j^{\backslash 2,j} \left( \left(\tilde{V}_{jj}\right)^{-1} \tilde{m}_j - \tilde{v}_{2,j}^{-1} \tilde{m}_{2,j} \right), \tag{15}$$

$$\gamma^{\backslash 2,j} = d\left( \tilde{\gamma}_j, \tilde{\gamma}_{2,j} \right) \tag{16}$$

**Moment matching for the second term**

Computing the normalization for $f_{2,j} Q^{\backslash 2,j}$:

$$Z_{2,j} = \sum_{\boldsymbol{z}} \int \int f_{2,j}(x_j, z_j) Q^{\backslash 2,j}\left(\boldsymbol{x}, \boldsymbol{z}, \boldsymbol{\gamma}\right) \mathrm{d}\boldsymbol{\gamma} \mathrm{d}\boldsymbol{x}$$

$$= \sum_{z_j} \int f_{2,j}(x_j, z_j) \sum_{\boldsymbol{z}_{\backslash j}} \int \int Q^{\backslash 2,j}\left(\boldsymbol{x}, \boldsymbol{z}, \boldsymbol{\gamma}\right) \mathrm{d}\boldsymbol{\gamma} \mathrm{d}\boldsymbol{x}_{\backslash j} \mathrm{d}x_j \tag{17}$$

Plugging in the densities

$$Z_{2,j} = \sum_{z_j} \int \left[(1 - z_j)\delta(x_j) + z_j \mathcal{N}\left(x_j | 0, \tau\right)\right] \sum_{\boldsymbol{z}_{\backslash j}} \int \int \mathcal{N}\left(\boldsymbol{x} | \boldsymbol{m}^{\backslash 2,j}, \boldsymbol{V}^{\backslash 2,j}\right) \prod_{i=1}^{D} \mathrm{Ber}\left(z_i | \phi\left(\gamma_i^{\backslash 2,j}\right)\right) \mathcal{N}\left(\boldsymbol{\gamma} | \tilde{\mu}, \tilde{\Sigma}\right) \mathrm{d}\boldsymbol{\gamma} \mathrm{d}\boldsymbol{x}_{\backslash j}$$

$$= \sum_{z_j} \int \left[(1 - z_j)\delta(x_j) + z_j \mathcal{N}\left(x_j | 0, \tau\right)\right] \mathrm{Ber}\left(z_j | \phi\left(\gamma_j^{\backslash 2,j}\right)\right) \mathcal{N}\left(x_j | m_j^{\backslash 2,j}, v_j^{\backslash 2,j}\right) \mathrm{d}x_j$$

$$= \left(1 - \phi\left(\gamma_j^{\backslash 2,j}\right)\right) \mathcal{N}\left(0 | m_j^{\backslash 2,j}, v_{jj}^{\backslash 2,j}\right) + \phi\left(\gamma_j^{\backslash 2,j}\right) \mathcal{N}\left(x_j | m_j^{\backslash 2,j}, \tau + v_j^{\backslash 2,j}\right) \tag{18}$$

For the moment matching, we need the following moments:

$$\mathbb{E}_{f_{2,j}Q^{\backslash 2,j}}\left[\boldsymbol{x}\right],\quad \mathbb{E}_{f_{2,j}Q^{\backslash 2,j}}\left[\boldsymbol{x}\boldsymbol{x}^T\right],\quad \&\quad \mathbb{E}_{f_{2,j}Q^{\backslash 2,j}}\left[\boldsymbol{z}\right] \tag{19}$$

The moment matching results in the following update equations:

$$v_{jj}^{\text{new}} = \frac{a_j}{a_j+b_j}\left[\frac{\left(m_j^{\backslash 2,j}\tau\right)^2}{\left(v_j^{\backslash 2,j}+\tau\right)^2}+\frac{v_j^{\backslash 2,j}\tau}{v_j^{\backslash 2,j}+\tau}\right]-\left[\frac{a_j}{a_j+b_j}\frac{m_j^{\backslash 2,j}\tau}{v_j^{\backslash 2,j}+\tau}\right]^2$$

$$\tilde{v}_{2,j}^{\text{new}} = \left[\left(v_{jj}^{\text{new}}\right)^{-1}-\left(v_j^{\backslash 2,j}\right)^{-1}\right]^{-1} \tag{20}$$

$$m_j^{\text{new}} = \frac{a_j}{a_j+b_j}\frac{m_j^{\backslash 2,j}\tau}{v_j^{\backslash 2,j}+\tau} \tag{21}$$

$$\tilde{m}_{2,j}^{\text{new}} = v_{2,j}^{\text{new}}\left(\left(v_j^{\text{new}}\right)^{-1}m_j^{\text{new}}-\left(v_j^{\backslash 2,j}\right)^{-1}m_j^{\backslash 2,j}\right) \tag{22}$$

$$\tilde{\gamma}_j^{\text{new}} = \phi^{-1}\left(\frac{a_j}{a_j+b_j}\right) \tag{23}$$

$$\tilde{\gamma}_j^{\text{new}} = d\left(\tilde{\gamma}_j^{\text{new}},\gamma_j^{\backslash 2,j}\right) \tag{24}$$

where

$$a_j = \phi\left(\gamma_j^{\backslash 2,j}\right)\mathcal{N}\left(0\big|m_j^{\backslash 2,j},\tau+v_j^{\backslash 2,j}\right) \tag{25}$$

$$b_j = \left(1-\phi\left(\gamma_j^{\backslash 2,j}\right)\right)\mathcal{N}\left(0\big|m_j^{\backslash 2,j},v_{jj}^{\backslash 2,j}\right) \tag{26}$$

**Computing the cavity distributions for the third term**

$$Q^{\backslash 3,j}(\boldsymbol{x},\boldsymbol{z},\boldsymbol{\gamma}) = \frac{Q(\boldsymbol{x},\boldsymbol{z},\boldsymbol{\gamma})}{f_{3,j}(z_j,\gamma_j)} = \frac{\mathcal{N}\left(\boldsymbol{x}\big|\tilde{\boldsymbol{m}},\tilde{\boldsymbol{V}}\right)\prod_{i=1}^{D}\text{Ber}\left(z_i\big|\phi\left(\tilde{\gamma}_i\right)\right)\mathcal{N}\left(\boldsymbol{\gamma}\big|\tilde{\boldsymbol{\mu}},\tilde{\boldsymbol{\Sigma}}\right)}{\text{Ber}\left(z_j\big|\phi\left(\tilde{\gamma}_{3,j}\right)\right)\mathcal{N}\left(\gamma_j\big|\tilde{\mu}_{3,j},\tilde{\Sigma}_{3,j}\right)} \tag{27}$$

The parameters for the $j$'th marginal cavity distribution then becomes

$$\Sigma_j^{\backslash 3,j} = \left(\left(\tilde{V}_{jj}\right)^{-1}-\left(\tilde{\Sigma}_{3,j}\right)^{-1}\right)^{-1} \tag{28}$$

$$\mu_j^{\backslash 3,j} = \Sigma_j^{\backslash 3,j}\left(\left(\tilde{V}_{jj}\right)^{-1}\tilde{m}_j-\left(\tilde{\Sigma}_{3,j}\right)^{-1}\tilde{\mu}_{3,j}\right), \tag{29}$$

$$\gamma_j^{\backslash 3,j} = d\left(\tilde{\gamma}_j,\tilde{\gamma}_{3,j}\right) \tag{30}$$

**Moment matching for the third term**

We need to following moments

$$\mathbb{E}_{f_{3,j}Q^{\backslash 3,j}}\left[\boldsymbol{\gamma}\right],\quad \mathbb{E}_{f_{3,j}Q^{\backslash 3,j}}\left[\boldsymbol{\gamma}\boldsymbol{\gamma}^T\right],\quad \&\quad \phi\left(\boldsymbol{\gamma}\right)^{\text{new}}=\mathbb{E}_{f_{3,j}Q^{\backslash 3,j}}\left[\boldsymbol{z}\right] \tag{31}$$

Computing the normalization for $f_{3,j}Q^{\backslash 3,j}$

$$Z_{3,j} = \sum_{\boldsymbol{z}}\int\int f_{3,j}(z_j,\gamma_j)Q^{\backslash 3,j}\left(\boldsymbol{x},\boldsymbol{z},\boldsymbol{\gamma}\right)\mathrm{d}\boldsymbol{\gamma}\mathrm{d}\boldsymbol{x}$$

$$= \sum_{z_j}\int f_{3,j}(z_j,\gamma_j)\sum_{\boldsymbol{z}_{\backslash j}}\int\int Q^{\backslash 3,j}\left(\boldsymbol{x},\boldsymbol{z},\boldsymbol{\gamma}\right)\mathrm{d}\boldsymbol{x}\mathrm{d}\boldsymbol{\gamma}_{\backslash j}\mathrm{d}\gamma_j$$

$$= \sum_{z_j}\int f_{3,j}(z_j,\gamma_j)Q^{\backslash 3,j}\left(z_j,\gamma_j\right)\mathrm{d}\gamma_j \tag{32}$$

where $\boldsymbol{x}, \boldsymbol{z}_{\backslash j}$ and $\boldsymbol{\gamma}_{\backslash j}$ are marginalized out. Plugging in the densities

$$Z_{3,j} = \sum_{z_j} \int Ber\left(z_j | \phi\left(\gamma_j\right)\right) Ber\left(z_j | \phi\left(\gamma_j^{\backslash 3,j}\right)\right) \mathcal{N}\left(\gamma_j | \mu_j^{\backslash 3,j}, \boldsymbol{\Sigma}_j^{\backslash 3,j}\right) \mathrm{d}\gamma_j \tag{33}$$

Carrying out the sum

$$Z_{3,j} = \int \left[\left(1 - \phi\left(\gamma_j\right)\right)\left(1 - \phi\left(\gamma_j^{\backslash 3,j}\right)\right) + \phi\left(\gamma_j\right)\phi\left(\gamma_j^{\backslash 3,j}\right)\right] \mathcal{N}\left(\gamma_j | \mu_j^{\backslash 3,j}, \boldsymbol{\Sigma}_j^{\backslash 3,j}\right) \mathrm{d}\gamma_j \tag{34}$$

Applying linearity of integrals

$$Z_{3,j} = \left(1 - \phi\left(\gamma_j^{\backslash 3,j}\right)\right) \int \left(1 - \phi\left(\gamma_j\right)\right) \mathcal{N}\left(\gamma_j | \mu_j^{\backslash 3,j}, \boldsymbol{\Sigma}_j^{\backslash 3,j}\right) \mathrm{d}\gamma_j$$
$$+ \phi\left(\gamma_j^{\backslash 3,j}\right) \int \phi\left(\gamma_j\right) \mathcal{N}\left(\gamma_j | \mu_j^{\backslash 3,j}, \boldsymbol{\Sigma}_j^{\backslash 3,j}\right) \mathrm{d}\gamma_j \tag{35}$$

The integral evaluate to

$$\int \phi\left(\gamma_j\right) \mathcal{N}\left(\gamma_j | \mu_j^{\backslash 3,j}, \boldsymbol{\Sigma}_j^{\backslash 3,j}\right) \mathrm{d}\gamma_j = \phi\left(\frac{\mu_j^{\backslash 3,j}}{\sqrt{1 + \boldsymbol{\Sigma}_j^{\backslash 3,j}}}\right) = \phi(z) \equiv C_{3,j} \tag{36}$$

where we have defined $z$ as

$$z = \frac{\mu_j^{\backslash 3,j}}{\sqrt{1 + \boldsymbol{\Sigma}_j^{\backslash 3,j}}} \tag{37}$$

Therefore, the normalization becomes

$$Z_{3,j} = \left(1 - \phi\left(\gamma_j^{\backslash 3,j}\right)\right)\left(1 - C_{3,j}\right) + \phi\left(\gamma_j^{\backslash 3,j}\right) C_{3,j} \tag{38}$$

Similarly, we compute the first moment of $\boldsymbol{z}$

$$\phi\left(\gamma_j\right)^{\text{new}} = \frac{1}{Z_{3,j}} \sum_{z_j} \int z_j \cdot f_{3,j}(z_j, \gamma_j) Q^{\backslash 3,j}\left(z_j, \gamma_j\right) \mathrm{d}\gamma_j$$
$$= \frac{1}{Z_{3,j}} \sum_{z_j} \int z_j Ber\left(z_j | \phi\left(\gamma_j\right)\right) Ber\left(z_j | \phi\left(\gamma_j^{\backslash 3,j}\right)\right) \mathcal{N}\left(\gamma_j | \mu_j^{\backslash 3,j}, \boldsymbol{\Sigma}_j^{\backslash 3,j}\right) \mathrm{d}\gamma_j$$
$$= \frac{1}{Z_{3,j}} \phi\left(\gamma_j^{\backslash 3,j}\right) \int \phi\left(\gamma_j\right) \mathcal{N}\left(\gamma_j | \mu_j^{\backslash 3,j}, \boldsymbol{\Sigma}_j^{\backslash 3,j}\right) \mathrm{d}\gamma_j$$
$$= \frac{1}{Z_{3,j}} \phi\left(\gamma_j^{\backslash 3,j}\right) C_{3,j} \tag{39}$$

where the result in eq. (36) is used. Therefore

$$\gamma_j^{\text{new}} = \phi^{-1}\left(\frac{1}{Z_{3,j}} \phi\left(\gamma_j^{\backslash 3,j}\right) C_{3,j}\right) \tag{40}$$

Hence, the update becomes

$$\tilde{\gamma}_{3,j} = d\left(\gamma_j^{\text{new}}, \gamma_j^{\backslash 3,j}\right) \tag{41}$$

Similarly, the first moment w.r.t. $\gamma_j$ evaluates to

$$\mu_j^{\text{new}} = \frac{1}{Z_{3,j}} \sum_{z_j} \int \gamma_j Ber\left(z_j | \phi\left(\gamma_j\right)\right) Ber\left(z_j | \phi\left(\gamma_j^{\backslash 3,j}\right)\right) \mathcal{N}\left(\gamma_j | \mu_j^{\backslash 3,j}, \boldsymbol{\Sigma}_j^{\backslash 3,j}\right) \mathrm{d}\gamma_j$$
$$= \frac{1}{Z_{3,j}} \int \gamma_j \left[\left(1 - \phi\left(\gamma_j\right)\right)\left(1 - \phi\left(\gamma_j^{\backslash 3,j}\right)\right) + \phi\left(\gamma_j\right)\phi\left(\gamma_j^{\backslash 3,j}\right)\right] \mathcal{N}\left(\gamma_j | \mu_j^{\backslash 3,j}, \boldsymbol{\Sigma}_j^{\backslash 3,j}\right) \mathrm{d}\gamma_j$$
$$= \frac{1}{Z_{3,j}} \left(1 - \phi\left(\gamma_j^{\backslash 3,j}\right)\right) \int \gamma_j \left(1 - \phi\left(\gamma_j\right)\right) \mathcal{N}\left(\gamma_j | \mu_j^{\backslash 3,j}, \boldsymbol{\Sigma}_j^{\backslash 3,j}\right) \mathrm{d}\gamma_j$$
$$+ \frac{1}{Z_{3,j}} \phi\left(\gamma_j^{\backslash 3,j}\right) \int \gamma_j \phi\left(\gamma_j\right) \mathcal{N}\left(\gamma_j | \mu_j^{\backslash 3,j}, \boldsymbol{\Sigma}_j^{\backslash 3,j}\right) \mathrm{d}\gamma_j \tag{42}$$

Using the the results from ch. 3.9 in the Gaussian process book (www.gpml.org), we have the following result:

$$\int \gamma_j \phi\left(\gamma_j\right) \mathcal{N}\left(\gamma_j | \mu_j^{\backslash 3,j}, \mathbf{\Sigma}_j^{\backslash 3,j}\right) d\gamma_j = C_{3,j} \left( \mu_j^{\backslash 3,j} + \frac{\mathbf{\Sigma}_j^{\backslash 3,j} \mathcal{N}\left(z | 0, 1\right)}{C_{3,j} \sqrt{1 + \mathbf{\Sigma}_j^{\backslash 3,j}}} \right) \tag{43}$$

Note, that the scaling $C_{3,j}$ is necessary since the integral is not normalized. Now define

$$W_{3,j} = C_{3,j} \mu_j^{\backslash 3,j} + \frac{\mathbf{\Sigma}_j^{\backslash 3,j} \mathcal{N}\left(z | 0, 1\right)}{\sqrt{1 + \mathbf{\Sigma}_j^{\backslash 3,j}}} \tag{44}$$

and inserting this result yields the expression for the first moment

$$\mu_j^{\text{new}} = \frac{1}{Z_{3,j}} \left[ \left(1 - \phi\left(\gamma_j^{\backslash 3,j}\right)\right) \left[\mu_j^{\backslash 3,j} - W_{3,j}\right] + \phi\left(\gamma_j^{\backslash 3,j}\right) W_{3,j} \right] \tag{45}$$

And for the second moment, we get

$$\mathbb{E}_{f_{3,j} Q^{\backslash 3,j}}\left[\gamma_j^2\right] = \frac{1}{Z_{3,j}} \sum_{z_j} \int \gamma_j^2 \text{Ber}\left(z_j | \phi\left(\gamma_j\right)\right) \text{Ber}\left(z_j | \phi\left(\gamma_j^{\backslash 3,j}\right)\right) \mathcal{N}\left(\gamma_j | \mu_j^{\backslash 3,j}, \mathbf{\Sigma}_j^{\backslash 3,j}\right) d\gamma_j$$

$$= \frac{1}{Z_{3,j}} \left(1 - \phi\left(\gamma_j^{\backslash 3,j}\right)\right) \left[ \int \gamma_j^2 \mathcal{N}\left(\gamma_j | \mu_j^{\backslash 3,j}, \mathbf{\Sigma}_j^{\backslash 3,j}\right) d\gamma_j - \int \gamma_j^2 \phi\left(\gamma_j\right) \mathcal{N}\left(\gamma_j | \mu_j^{\backslash 3,j}, \mathbf{\Sigma}_j^{\backslash 3,j}\right) d\gamma_j \right]$$

$$+ \frac{1}{Z_{3,j}} \phi\left(\gamma_j^{\backslash 3,j}\right) \int \gamma_j^2 \phi\left(\gamma_j\right) \mathcal{N}\left(\gamma_j | \mu_j^{\backslash 3,j}, \mathbf{\Sigma}_j^{\backslash 3,j}\right) d\gamma_j \tag{46}$$

Using the same result for the integral, we get the following result

$$\int \gamma_j^2 \phi\left(\gamma_j\right) \mathcal{N}\left(\gamma_j | \mu_j^{\backslash 3,j}, \mathbf{\Sigma}_j^{\backslash 3,j}\right) d\gamma_j = 2\mu_j^{\backslash 3,j} W_{3,j} + C_{3,j} \left[ \mathbf{\Sigma}_j^{\backslash 3,j} - \left(\mu_j^{\backslash 3,j}\right)^2 \right] - \frac{\left(\mathbf{\Sigma}_j^{\backslash 3,j}\right)^2 z \mathcal{N}\left(z | 0, 1\right)}{\left(1 + \mathbf{\Sigma}_j^{\backslash 3,j}\right)} \equiv M_{3,j}$$

Inserting $M_{3,j}$:

$$\mathbb{E}_{f_{3,j} Q^{\backslash 3,j}}\left[\gamma_j^2\right] = \frac{1}{Z_{3,j}} \left[ \left(1 - \phi\left(\gamma_j^{\backslash 3,j}\right)\right) \left[\left(\mu_j^{\backslash 3,j}\right)^2 + \mathbf{\Sigma}_j^{\backslash 3,j} - M_{3,j}\right] + \phi\left(\gamma_j^{\backslash 3,j}\right) M_{3,j} \right] \tag{47}$$

Finally, we obtain the variance as:

$$\Sigma_j^{\text{new}} = \mathbb{E}_{f_{3,j} Q^{\backslash 3,j}}\left[\gamma_j^2\right] - \mathbb{E}_{f_{3,j} Q^{\backslash 3,j}}\left[\gamma_j\right]^2 \tag{48}$$

The update equations for the mean and variance are then given by:

$$\tilde{\Sigma}_{3,j}^{\text{new}} = \left[ \left(\Sigma_j^{\text{new}}\right)^{-1} - \left(\Sigma^{\backslash 3,j}\right)^{-1} \right]^{-1} \tag{49}$$

$$\tilde{\mu}_{3,j}^{\text{new}} = \tilde{\Sigma}_{3,j}^{\text{new}} \left( \left(\Sigma_j^{\text{new}}\right)^{-1} \mu_j^{\text{new}} - \left(\Sigma^{\backslash 3,j}\right)^{-1} \mu_j^{\backslash 3,j} \right) \tag{50}$$

(a) Spectrum     (b) NMSE     (c) F-measure

(d) Run time     (e) Iterations

Figure 1: Illustration of the properties of the low rank approximation of $\boldsymbol{\Sigma}_0$. Data are generated the way as described for experiment 1 in the paper, except that $\boldsymbol{A} \in \mathbb{R}^{125 \times 500}$ and the sparsity level is fixed at $K/D = 0.1$.

### Experiment: Effect of low rank approximation

This experiment is designed to investigate the properties and implications of the low rank approximation of $\boldsymbol{\Sigma}_0$. We generate the problems in the same ways as in the first experiment (described in the paper), but now sweep over the rank $R$ of the approximation of $\boldsymbol{\Sigma}_0$ with fixed problem size, i.e. $N = 125$ and $D = 500$. The results are shown in figure 1. For the specific choice of covariance function, figure 1(b)-(c) shows that a 40-rank approximation does not introduce significant errors in terms of NMSE and F-measure. but the run time is reduced by a factor $\approx 3.5$. The reduction is expected to become even more significant as $D$ increases.

### Experiment: Shepp Logan Recovery

We have also recreated the Shepp-Logan Phantom experiment from [1] with $D = 10^4$ unknowns, $K = 1723$ non-zero weights, $N = 2K$ observations and SNR $= 10dB$. That is, we generated a set of measurements using the model $\boldsymbol{y} = \boldsymbol{A}\boldsymbol{x}_0 + \boldsymbol{e}$, where the true signal $\boldsymbol{x}_0$ is the Shepp-Logan Phantom image (see figure 2a). For the EP method, we imposed a squared exponential covariance function with length-scale 8 for $\boldsymbol{\gamma}$ defined on the $100 \times 100$ image grid. We use three methods for reconstruction $\boldsymbol{x}_0$: Our proposed method, BG-AMP [2] and an oracle estimator, which computes a least squares estimate based on knowledge of the true support. We consider the Normalized Mean Square Error (NMSE) of the estimated coefficients $\hat{\boldsymbol{x}}$ as well as the F-measure of the estimated support $\hat{z}$. The reconstructions are shown in figure 2, where the first row shows the reconstructed coefficients and the second row shows the reconstructed support. Our proposed method yields $F_{sq} = 0.994$ and NMSE$_{sq}$ $= 0.336$ for this experiment, whereas BG-AMP yields $F = 0.624$ and NMSE $= 0.717$. For reference, the oracle estimator yields NMSE $= 0.326$.

(a) True coefficients $\boldsymbol{x}_0$     (b) $\hat{\boldsymbol{x}}_{\text{EP}}$     (c) $\hat{\boldsymbol{x}}_{\text{BG-AMP}}$     (d) $\hat{\boldsymbol{x}}_{\text{oracle}}$

(e) True support $\boldsymbol{z}_0$     (f) $\hat{\boldsymbol{z}}_{\text{EP}}$     (g) $\hat{\boldsymbol{z}}_{\text{BG-AMP}}$     (h) $\hat{\boldsymbol{z}}_{\text{oracle}}$

Figure 2: Recovery of the Shepp-Logan Phantom. The first row shows the reconstructed coefficients $\hat{x}$ and the second row shows the reconstructed support $\hat{z}$.