[Reviews · NeurIPS 2014]

Submitted by Assigned_Reviewer_4

The paper presents a simple novel model for structured sparsity in spike-and-slab models and an expectation propagation algorithm for Bayesian inference. It is written clearly and accompanied by interesting examples and comparisons with other approaches.
Summary: Although there are some previous works in this area, this particular approach is simple and novel.

Submitted by Assigned_Reviewer_13

This paper introduces a new spike and slab sparse prior, where the sparse variables are allowed to be correlated and this structure is encoded by generic covariance functions. Therefore, this is a new spike and slab prior for modeling structured sparsity patterns. The authors also propose an algorithm for inference of spike and slab variables under the proposed prior, by applying the framework of expectation propagation. Approximations to a covariance matrix inverse are also proposed, in order to reduce the computational complexity of the algorithm. Numerical results demonstrate that the method significantly outperforms prior art for problems of sparse recovery that exhibit such form of structured sparsity.

This paper is an original contribution to the field of sparse recovery and proposes a solution to a significant problem of inference for structured sparsity, one that has been a challenge for many researchers in the last decade. Although the paper does not solve the problem for any general joint distribution of sparse variables, it proposes a solution for a large set of distributions. Therefore, I think it will be of interest for a larger audience and probably useful to many researchers in the field.

The paper is clearly written, though notation can be improved (see below for detailed suggestions). Acronyms are not always defined, for example MMV.

One improvement I would like to see in the revised paper (besides detailed comments below) is the addition of real data examples where the structure in the sparse variables can be modeled by the set of distributions considered in the paper. The example given in the current version, with EEG measurement matrix, is not convincing since the sparse variables are still sampled from a distribution defined by the authors. I would not call it a real example, since the only real data is the measurement matrix and its properties are not really the target of the paper. Having such examples would make the paper much stronger and the proposed method would have a larger impact.

Another point that would be good to discuss in the paper is the attenuation of correlation of sparse variables with respect to the correlation of parameters gamma (Fig. 1c). What does the attenuation depend on? Is there a way to control that dependence? How does the curvature of curves in Fig.1c change with other parameters? I believe this will be an important point to consider in real applications.

Detailed comments:
- Sec. 2 title: typo - should be "slab" instead of "slap"
- line 093: "easy to generate" instead of "easy the generate"
- Sec.3. When replacing f1 with f1 tilde, it would be good to explain the switch from the conditional distribution of y given x to a conditional distribution of x given m1 and V1 (where m1 depends on y)
- Sec.3. It is not clear where the function d(x,y) is used in 18-21
- Sec.3.1. Notation tilde(f)_a,i should be explained, previously was only used tilde(f)_a and now another subscript was added. Maybe it would be better to put i in the superscript? Or just define it in the text
- Using bold A and regular A for two different variables is confusing, please consider modifying one.
- Sec. 3.2: What is N? Rank of V1?
- line 306: "slab" instead of "slap"
Summary: Overall, this is an interesting, technically sound paper that proposes a new structured sparsity prior and provides an algorithm for inference under this prior. Numerical results demonstrate the power of the proposed method in structured sparse recovery.

Submitted by Assigned_Reviewer_36

This paper studies a structured spike-and-slab prior where Gaussian process is used to impose spatial contiguity of the sparsity pattern. Previous work in (Cevher et. al., 2008) reported a closely related study by using Ising model to encode the between-element correlation. A major difference between the present paper and (Cevher et. al., 2008) is that the Ising model is replaced by a Gaussian process passed through a Probit link function. The proposed spike-and-slab construction is employed as a prior for the unknown signal in a linear inverse problem, and an EP algorithm is developed to infer the unknown signal under such a prior. Experiments are based on synthetic data.

The Gaussian process based correlation structure constitutes the main novelty of the proposed spike-and-slab prior. However, how this correlation structure compares to the Ising model in (Cevher et. al., 2008) is not discussed, nor are they compared in the experiments. In addition to this obvious issue, a number of other issues are identified, as described below.

(1) Figure 1(c), how is the correlation between z1 and z2 computed?

(2) The shorthand notation MMV is never defined.

(3) No real data are used in the experiments.

(4) Experiment 2, why not including BG-AMP in the comparison, although (quoted from the paper) ''Note, that the BG-AMP algorithm is expected to perform poorly on this problem due to the violation of the assumption of Aij being gaussian iid."

Summary: This paper presents a correlated spike-and-slab prior with the correlation structure implemented by a Gaussian process. The relations between the proposed prior and the Ising-model based prior in (Cevher et. al., 2008) are not sufficiently discussed, nor are they compared in the experiments. Other issues include the missing comparison to BG-AMP in experiment 2 and that the experiments are based exclusively on synthetic data.
Author Feedback
Author rebuttal: Thank you for the very useful feedback.

Our proposed model is similar to the Markov Random Field (MRF) prior presented in (Cevher et al, 2008) as Reviewer ID 36 points out. This specific paper is therefore also cited in our paper, but we should have been more specific about the similarities and differences:

- In contrast to the MRF prior, the Gaussian Process formulation in our model allows us to rapidly generate samples from the prior without having to resort to Gibbs sampling or similar.

- In our formulation the correlation structure is specified using generic covariance functions rather than indirectly through clique potentials.

- (Cevher et al, 2008) provides an algorithm for MAP-inference, whereas we provide a scheme for full Bayesian inference using expectation propagation.

- We are also considering more complex sensing matrices, i.e. as in the EEG example, rather sample simple i.i.d. Gaussian matrices.

We have reconstructed the Shepp-Logan phantom experiment from (Cevher et al, 2008) as accurately as possible solely based on their paper. Using our method we were able to reconstruct the true support of the phantom with an F measure of 0.994.

Furthermore, in a revised version of the paper, we would also like to include two more relevant references:

- Van Gerven, M., Cseke, B., Heskes, T., Van Gerven, M., Oostenveld, R., & Heskes, T. (2009). Bayesian source localization with the multivariate laplace prior. Advances in Neural Information Processing Systems 22 - Proceedings of the 2009 Conference, 1901-1909.

- Yu, L., Sun, H., Barbot, J. P., & Zheng, G. (2012). Bayesian compressive sensing for cluster structured sparse signals. SIGNAL PROCESSING, 92(1), 259-269. doi:10.1016/j.sigpro.2011.07.015

Finally, the typos and notational issues pointed out by the reviewers will be fixed right away.